# Characterizing SARS-CoV-2 neutralization profiles after bivalent boosting using antigenic cartography

Annika Rössler [1,4], Antonia Netzl [2,4], Ludwig Knabl[3], David Bante [1], Samuel H. Wilks [2], Wegene Borena[1], Dorothee von Laer [1], Derek J. Smith [2] ✉ & Janine Kimpel [1] ✉

Since emergence of the initial SARS-CoV-2 BA.1, BA.2 and BA.5 variants, Omicron has diversified substantially. Antigenic characterization of these new variants is important to analyze their potential immune escape from population immunity and implications for future vaccine composition. Here, we describe an antigenic map based on human single-exposure sera and live-virus isolates that includes a broad selection of recently emerged Omicron variants such as BA.2.75, BF.7, BQ, XBB and XBF variants. Recent Omicron variants clustered around BA.1 and BA.5 with some variants further extending the antigenic space. Based on this antigenic map we constructed antibody landscapes to describe neutralization profiles after booster immunization with bivalent mRNA vaccines based on ancestral virus and either BA.1 or BA.4/5. Immune escape of BA.2.75, BQ, XBB and XBF variants was also evident in bivalently boosted individuals, however, cross-neutralization was improved for those with hybrid immunity. Our results indicate that future vaccine updates are needed to induce cross-neutralizing antibodies against currently circulating variants.

In the past 3 years, a large variety of different variants of the severe acute respiratory syndrome coronavirus 2 (SARS-CoV-2) have emerged. Between the end of 2021 and the middle of 2022, the Omicron variants BA.1, BA.2, and BA.5 subsequently emerged and globally replaced pre-Omicron variants. Omicron variants have considerably more mutations in the spike protein compared to previously circulating variants, allowing the distinction into pre-Omicron and Omicron variants[1,2]. The accumulation of mutations resulted in a substantial escape of neutralizing antibody responses induced by previous infection with a pre-Omicron variant or vaccination[3–6]. However, a single infection with one of these early Omicron variants also induced only limited cross-neutralizing antibody responses against the other Omicron and pre-Omicron variants[7–9]. We and others have analyzed antigenic relations between these early Omicron and pre-Omicron variants

using antigenic cartography. On antigenic maps, pre-Omicron and Omicron variants are located distant from each other, but BA.1, BA.2, and BA.5 Omicron variants also differ considerably from each other[7,8,10,11].

The strong escape of Omicron variants from ancestral variant vaccinated sera suggested the need for adapted vaccines. Regularly updating vaccines to contain currently circulating variants is long known for influenza, where the composition of the next vaccine is selected annually after the flu season based on antigenic characterization of circulating strains. In June 2022, the Technical Advisory Group on COVID-19 Vaccine Composition (TAG-CO-VAC) by the World Health Organization (WHO) recommended the inclusion of an Omicron variant in updated COVID-19 vaccines[12]. Both BioNTech/Pfizer and Moderna developed updated bivalent mRNA vaccines based on

[1]Institute of Virology, Department of Hygiene, Microbiology and Public Health, Medical University of Innsbruck, Peter-Mayr-Str. 4b, 6020 Innsbruck, Austria. [2]University of Cambridge, Centre for Pathogen Evolution, Department of Zoology, Cambridge, UK. [3]Tyrolpath Obrist Brunhuber GmbH, Hauptplatz 4, 6511 Zams, Austria. [4]These authors contributed equally: Annika Rössler, Antonia Netzl. ✉e-mail: djs200@cam.ac.uk; Janine.Kimpel@i-med.ac.at

ancestral virus and either BA.1 Omicron (BA.1 biv.) or BA.4/5 Omicron (BA.4/5 biv.), and these new vaccines were approved in many countries for booster vaccination. Clinical trials showed that a fourth dose with an updated vaccine boosted neutralizing antibodies against Omicron variants better than a booster with the original vaccine[13–16]. Studies also showed improved neutralization of recent Omicron variants (BA.2.75, XBB lineages) after a bivalent boost compared to an ancestral boost, however, the escape of these variants from bivalently boosted sera is still considerable[17–20]. By convergent evolution, many of these newly emerging Omicron variants from different lineages introduced mutations in spikes already associated with immune escape. Indeed some of these new variants escaped neutralization by some or all of the therapeutic monoclonal antibodies and were only weakly neutralized by sera from previously vaccinated or hybrid immune individuals[21].

Currently, sub-lineages of the BA.2.75, BA.5, and the recombinant XXB lineage circulate globally, with different lineages dominating in different countries. However, BA.1 and the parental BA.5, which are included in the updated vaccines, no longer play a role in overall infections. With new variants emerging, their antigenic characterization is an important question, also in light of evaluating the need for further vaccine updates. Some of these new variants already have been partially characterized regarding their neutralization profile, but most of these studies have been performed using sera of individuals after multiple exposures through infection and/or vaccination[18,19,21]. However, to clearly disentangle antigenic relations between virus variants, single exposure sera are most informative as antigenic maps represent titer differences among variants and are constructed based on fold drops from the maximum titer antigen in a serum (usually the infecting/vaccinating antigen) to other variant titers. We have previously shown that multiple exposures decrease these fold drops in titer between exposed and non-exposed variants[7]. Using multi-exposure sera to construct a map, therefore, may underestimate distances between variants due to higher cross-neutralization after multiple exposures rather than similar titers due to similar neutralization properties. Antibody landscapes, which are constructed to visualize neutralization profiles in a third dimension on top of a single exposure antigenic map, are a more appropriate tool to depict immunity after more complex exposure histories[22].

Additionally, many studies on recently emerged SARS-CoV-2 variants compare only a few virus variants, and a direct comparison across studies is often difficult due to different methodologies and virus strains. In the current study, we aimed to analyze antigenic relationships of a broad collection of pre-Omicron and Omicron variants, including BA.2.75, BA.5, XBB, and XBF recombinant lineages. Using this antigenic map as a basis, we subsequently generated antibody landscapes from BA.1 biv. or BA.4/5 biv. boosted individuals to characterize neutralization profiles elicited by these updated bivalent vaccines in the current antigenic space.

## Results

We have previously described an antigenic map containing major pre-Omicron variants as well as BA.1, BA.2, and BA.5 Omicron variants[7]. Meanwhile, many more Omicron variants have emerged. To characterize antigenic relations of these newly emerged Omicron variants relative to older variants, we aimed to isolate a representative set of variants from BA.2.75, BA.5, recombinant XBB, and recombinant XBF lineages (Supplementary Figs. 1–5). We generated virus stocks for three BA.2.75 variants (CB.1, BR.3, CH.1.1), six BA.5 variants (BA.5.2.1, BE.1.1, BF.7, BQ.1.3, BQ.1.1, BQ.1.18), two XBB recombinant variants (XBB.1, XBB.1.5.1), and one XBF recombinant variant (XBF.3) and confirmed their identity via third generation sequencing (Supplementary Table 2). Spike mutations for all variants used in this study relative to ancestral Wuhan-1 are shown in Supplementary Figs. 1–5.

In a second step, we characterized neutralization profiles for these new variants and determined their antigenic relation to early Omicron

as well as pre-Omicron variants. Antigenic relations can be visualized in an antigenic map, which is generated by translating fold changes of neutralization titers between variants into antigenic map distances[23]. To create antigenic maps that reflect the basic antigenic relationships among variants it is crucial to use single variant exposure sera, as multiple exposures increase cross-neutralization and therefore will potentially skew antigenic relations. We have earlier collected a number of first exposure sera for ancestral, alpha, beta, delta, BA.1 Omicron and BA.2 Omicron variants[7]. To increase resolution of an antigenic map in the area covered by Omicron variants, we included also one BA.5 first exposure serum and two CK.2.1.1 (BA.5.2 variant) first exposure sera in this study (Supplementary Fig. 21 for mutations in CK.2.1.1 spike compared to BA.5). We analyzed neutralizing antibodies for first infection sera and two dose BNT162b2 (BNT, Comirnaty, BioNTech/Pfizer) vaccinated individuals (BNT/BNT) against our panel of recently isolated Omicron variants as well as early BA.1, BA.2, and BA.5 Omicron variants and pre-Omicron variants (Fig. 1 and Supplementary Fig. 22). Three times BNT vaccinated individuals (BNT/BNT/BNT) were included as reference but the data from this cohort has not been used for calculation of the antigenic map as multi exposure sera likely underestimate antigenic relationships due to increased cross-reactivity and we previously showed that neutralization profiles of three dose vaccinated individuals were more similar to those of individuals after reinfection with an antigenically distinct variant than after single infection[7]. However, we constructed antibody landscapes for this group to visualize neutralization profiles (Supplementary Fig. 23).

Titers against the BA.2.75 variants CB.1, BR.3, and CH.1.1 were low or undetectable for most single-exposure sera indicating the strong immune escape phenotype of these variants (Fig. 1, purple box). BA.5 variants could be divided into two groups regarding their neutralization profiles. BA.5.2.1, BE.1.1, and BF.7 were more similar to the initial BA.5 variant, while BQ variants showed a greater drop in neutralizing antibodies (Fig. 1, petrol box). The three BQ variants analyzed in this study differed in spike mainly by presence of the 144 deletion and the R346T mutation. BQ.1.18 contains both, BQ.1.1 only the R346T mutation and BQ.1.3 neither (Supplementary Fig. 4). However, BQ.1.3 has an additional E619Q mutation. All three variants showed similar neutralization profiles except for neutralization by the CK.2.1.1 convalescent sera (Supplementary Fig. 21). While BQ.1.3 and BQ.1.18 were neutralized by both CK.2.1.1 sera, no neutralizing antibodies against BQ.1.1 were detected. In contrast, the single BA.5 convalescent sample did not contain neutralizing antibodies against any of the three analyzed BQ variants. Titers for both XBB recombinant variants dropped relative to earlier variants with a similar pattern for XBB.1 and XBB.1.5 (Fig. 1, orange box). The XBF.3 recombinant variant showed a similar neutralization pattern as the BA.2.75 variants, which was not unexpected as XBF.3 contains a BA.2.75 variant spike (Fig. 1, green box). Overall, although many of the new variants were poorly neutralized by single infection sera, most individuals vaccinated with three doses of BNT were able to neutralize the whole panel of analyzed variants at least at a low level.

To visualize the antigenic relation between virus variants, we next performed antigenic cartography[23] using data from the first infection and two dose vaccinated groups (see Supplementary Table 3 for sera included in the calculation of the antigenic map). In Fig. 2, we show our previously described map after addition of three BA.5/CK.2.1.1 sera and the panel of new variants[7]. In the antigenic map, colored circles indicate the position of the analyzed virus variants, and squares or triangles the position of the single variant exposure sera. Map proximity of viruses indicates a similar neutralization phenotype, and hence map distance reflects phenotypically distinct antigenic relationships. The positions of pre-Omicron and early Omicron variants BA.1, BA.2, and BA.5 did not change compared to our previous map[7]. Newly emerged Omicron variants clustered in the map area between and around BA.1 and BA.5, however also extended antigenic space substantially further

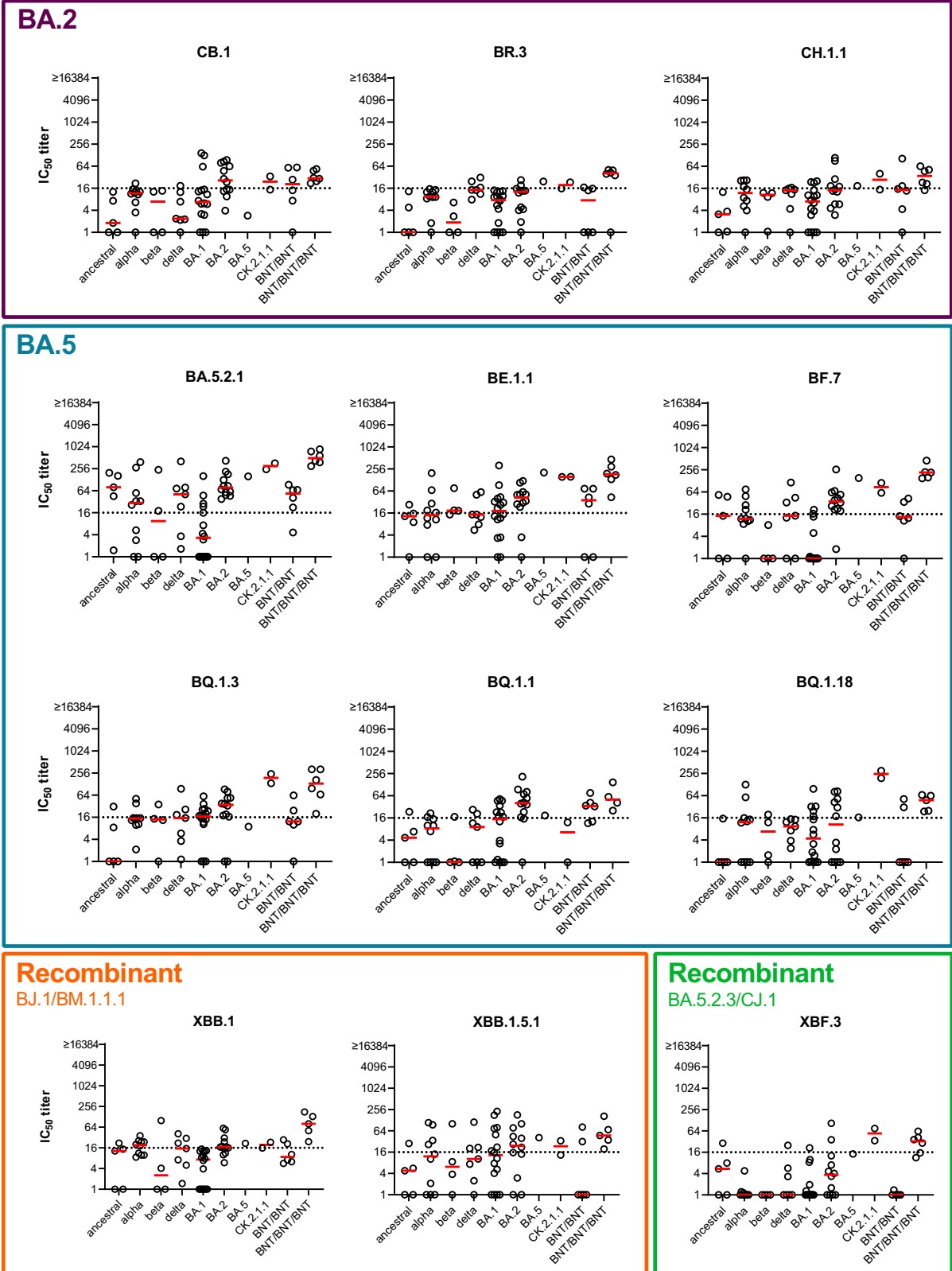

**Fig. 1 | Neutralization profiles of single variant exposure samples.** Single variant exposure plasma samples were collected, either from unvaccinated individuals after first infection with ancestral virus ($n = 5$), alpha ($n = 10$), beta ($n = 4$), delta ($n = 7$), BA.1 ($n = 16$), BA.2 ($n = 12$), BA.5 ($n = 1$), or CK.2.1.1 ($n = 2$) variant or from two (BNT/BNT, $n = 6$) or three-dose (BNT/BNT/BNT, $n = 6$) vaccinated (ancestral) individuals. Titers of neutralizing antibodies against BA.2.75 variants (CB.1, BR.3, CH.1.1), BA.5 variants (BA.5.2.1, BE.1.1, BF.7, BQ.1.3, BQ.1.1, BQ.1.18), recombinant XBB variants (XBB.1, XBB.1.5.1), or recombinant XBF.3 for individual patients (circles) and geometric mean (red line) are shown. Titers below 16 were treated as negative (dotted line), and titers below 1 were set to 1. BNT = BNT162b2, $IC_{50}$ titer = 50% neutralization titer.

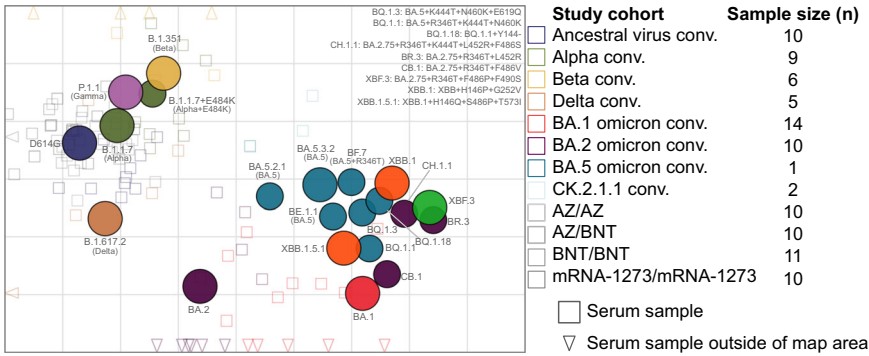

**Fig. 2 | Antigenic map constructed from human single exposure and double vaccination sera.** The antigenic map shows virus variants in colored circles and human sera as open squares in the color of their root variant or gray for vaccine sera and light blue for CK.2.1.1 sera. Triangles point in the direction of sera outside of the shown area (Supplement Fig. 6 for a non-zoomed in version). Each grid in the map corresponds to one twofold dilution of titers in the neutralization assay, making map distance a measure of antigenic similarity. Objects in the map are located relative to each other, *x*- and *y*-axis orientation is relative. Variants are labeled by pango lineage and colloquial name. For recent variants, spike substitutions are listed in the upper right of the map. The number of sera per cohort that was used to construct the map is shown in Supplementary Table 3.

from pre-Omicron variants. Most BA.5 variants are located close to our initial BA.5 isolate (BA.5.3.2), with BQ.1.1 being furthest away. The spike sequence identical BA.5 variants (BA.5.3.2, BA.5.2.1, and BE.1.1 with only an additional Q1208H mutation in S2) are within one antigenic unit, a distance which can be attributed to measurement noise (Supplementary Fig. 10). XBF.3 shows the greatest escape from pre-Omicron variants, and XBB.1 and XBB.1.5.1 occupy distinct positions from each other in the map due to slightly more escape of XBB.1 from BA.5 and CK.2.1.1 sera (Fig. 1). BA.2.75, XBB, and XBF variants, which all contain a BA.2 derived spike sequence, were positioned rather distant from BA.2 and further away from BA.2 than for example delta from BA.2. This can be explained by the low level of neutralization of these variants by sera from BA.2 convalescent individuals (Fig. 1). Consequently, exclusion of BA.2 convalescent sera moved BA.2.75 variants closer to BA.2 (Supplementary Fig. 12). Given the impact of sera in certain map areas to resolve this region of antigenic space, a limitation of the map is the low number of sera located in the area covered by the newer Omicron variants, increasing their position uncertainty compared to earlier variants (Supplementary Figs. 9–11 and 16). However, human first infection sera from these variants are extremely difficult to obtain after over three years of global SARS-CoV-2 circulation and vaccination campaigns.

As updated bivalent booster immunization contains either ancestral and BA.1 variant (BA.1 biv.) or ancestral and BA.4/5 variant (BA.4/5 biv.), we next investigated the level of cross-neutralizing antibodies in individuals who received three doses of ancestral virus vaccine followed by a fourth dose of one of the two bivalent boosters. We therefore collected plasma samples from individuals after a bivalent booster with or without previous infection history (Supplementary Table 1). In a first step, we analyzed antibody titers against the viral nucleocapsid (N) to detect previous infections. All four study participants with BA.4/5 biv. booster with a known history of infection (1 likely with alpha and 3 with BA.2 Omicron variant) were positive for N antibodies. Additionally, 16 of the participants without known infection history (5 in the BA.1 biv. boosted and 11 in the BA.4/5 biv. boosted group) were positive for N antibodies indicating a previous undetected infection (Supplementary Fig. 24 and Supplementary Table 1). Consequently, for these individuals the infecting variant is unfortunately unknown. Samples were grouped according to N antibody results in individuals with (BA.1 biv./N+ and BA.4/5 biv./N+) or without (BA.1 biv./N− and BA.4/5 biv./N−) previous infection. The interval between the last vaccine dose and blood collection had been approximately 1 month longer for the BA.1 biv. boosted groups compared to the BA.4/5 biv. boosted groups (Supplementary Table 1), which could influence

overall titers of neutralizing antibodies. Therefore, we first plotted titers against D614G, BA.1, and BA.5 across all four groups over time (Supplementary Fig. 25). Although neutralizing antibody titers tended to be lower for samples collected longer after immunization, no clear correlation was observed.

We further analyzed neutralizing antibody titers against a broader panel of variants, i.e., D614G, beta, delta, BA.1, BA.2, CB.1, BR.3, CH.1.1, BA.5 (BA.5.3.2), BF.7, BQ.1.3, BQ.1.1, BQ.1.18, XBB.1, XBB.1.5.1, and XBF.3 variants (Fig. 3 and Supplementary Fig. 26). For all bivalently boosted groups titers of neutralizing antibodies were in general higher compared to single exposure or three dose vaccinated cohorts analyzed in Fig. 1. Titers were especially high for pre-Omicron and early Omicron variants, but dropped against BA.2.75, BQ, XBB, and XBF variants. Interestingly, single individuals exhibited high neutralization titers against these variants as well. However, mean titers against these variants dropped ~4-fold or more against the reference variants D614G, BA.1, and BA.5 (Supplementary Fig. 26). For hybrid immune individuals, this drop was less pronounced and most individuals neutralized all analyzed variants. We next constructed antibody landscapes to better compare neutralization profiles between individuals with and without N antibodies (see Fig. 4 GMT landscapes and Supplementary Fig. 27 for individual landscapes). The different intervals between booster dose and blood collection again limited direct comparison between BA.1 biv. and BA.4/5 biv. boosted individuals. However, a comparison between N+ and N− shows that for both boosters the hybrid immunity landscapes were higher and flatter than the landscapes from N negative individuals indicating broader neutralization. The difference between hybrid and vaccine-only immunity was more pronounced in the BA.1 biv. groups. Timing between last vaccination and blood collection, which has been slightly longer for the BA.1 biv. groups compared to the BA.4/5 biv. groups could contribute to this difference.

Considering the almost complete escape from single exposure sera of the BQ, XBF, and BA.2.75 variants, their position in antigenic space could be even further away from pre-Omicron and early Omicron variants than in the current map[8,24]. As many titers against these variants were below the LOD (limit of detection), the actual titer difference is censored by the LOD. Consequently, a variant may be positioned at a distance corresponding to the fold drop from maximum titer to the LOD, however, it cannot accurately be estimated how much further away the variant actually is (Supplementary Figs. 10C and 16). This could only be resolved by increasing the resolution in this area of antigenic space through addition of first exposure sera from these variants, which are challenging to obtain

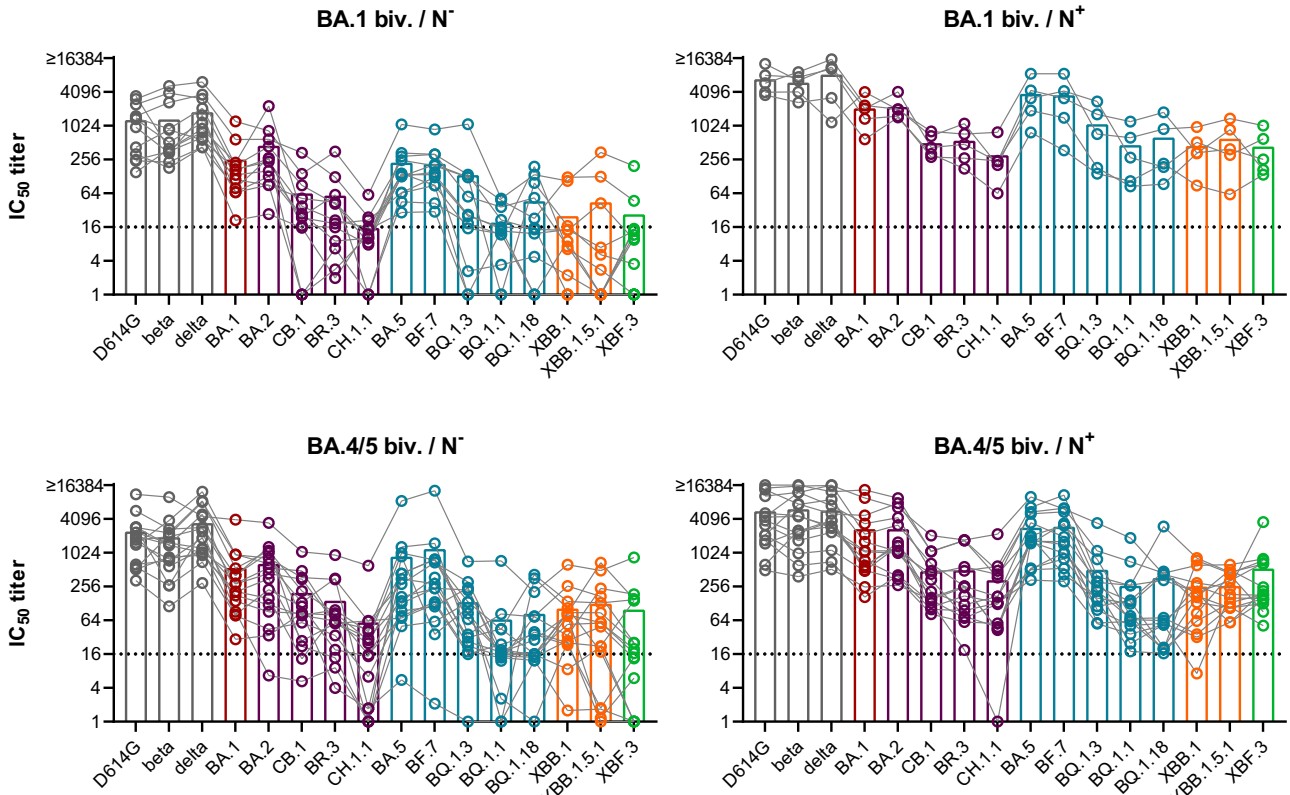

**Fig. 3 | Neutralization profiles after boosting with a forth dose of bivalent ancestral + BA.1 or ancestral + BA.4/5 vaccine.** Plasma was collected from individuals that received three doses of ancestral variant vaccine followed by a forth dose of either bivalent ancestral + BA.1 or ancestral + BA.4/5 vaccine. Antibodies against SARS-CoV-2 nucleocapsid (N) were determined using ELISA and samples were grouped accordingly: ancestral + BA.1 boost without detectable N antibodies (BA.1 biv./N⁻), $n = 12$; ancestral + BA.1 boost with positive N ELISA (BA.1 biv./N⁺), $n = 5$; ancestral + BA.4/5 boost without detectable N antibodies (BA.4/5 biv./N⁻), $n = 16$; ancestral + BA.4/5 boost with positive N ELISA (BA.4/5 biv./N⁺), $n = 15$. Titers of neutralizing antibodies against indicated variants are shown for individual patients as symbols connected by lines. Mean titers are shown as bars. Titers below 16 were treated as negative (dotted line), and titers below 1 were set to 1. IC$_{50}$ titer = 50% neutralization titer.

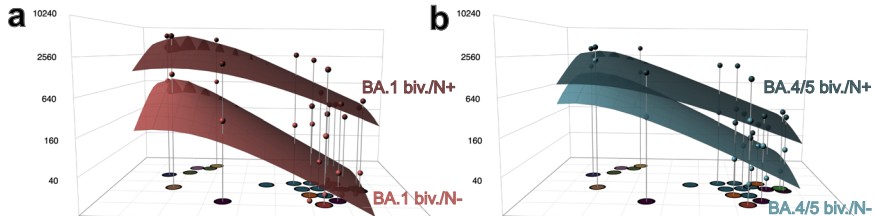

**Fig. 4 | Antibody landscapes after boosting with a fourth dose of bivalent ancestral + BA.1 or ancestral + BA.4/5 vaccine.** The map shown in Fig. 2 was used as a base map to construct antibody landscapes. Geometric mean titers (GMT) against each variant are shown on the *z*-axis above the corresponding position in the map. To construct landscapes, a continuous surface is fitted through these titers per individual. The geometric mean of these individual landscape fits is shown for **a** ancestral + BA.1 boost without detectable nucleocapsid (N) antibodies (BA.1 biv/N⁻, light red) and ancestral + BA.1 boost with positive N ELISA (BA.1 biv./N⁺, dark red) and **b** ancestral + BA.4/5 boost without detectable N antibodies (BA.4/5 biv./N⁻, light blue) and ancestral + BA.4/5 boost with positive N ELISA (BA.4/5 biv./N⁺, dark blue). The dots above each variant correspond to the GMT, which was calculated using the titertools R package[25].

given the current situation of population immunity. To test the impact of the underestimation of antigenic escape of the recent Omicron variants on the antibody landscapes, we constructed landscapes where the new variants were not included in the fitting procedure (Supplementary Fig. 28). All landscapes with fitting only pre-Omicron and early BA.1, BA.2 and BA.5 Omicron variants were flatter than when fitting all variants, and neutralization breadth differed only based on N antibody status but not on received booster. This observation suggests that some of the newer Omicron variants are indeed further out in antigenic space compared to their position in our current antigenic map.

## Discussion

The continuous emergence of new SARS-CoV-2 variants requires their antigenic characterization to assess their potential immune escape from current population immunity. Here, we described the neutralizing antibody response of human single infection and double vaccination sera against recently circulating SARS-CoV-2 variants using a live-virus approach to create an antigenic map, illustrating the variants' antigenic relationships in 2D. In our antigenic map, many recently emerging Omicron variants clustered in the area around and between BA.1 and BA.5, however, some variants occupied positions further out in antigenic space.

This was especially true for variants with BA.2.75-derived spike sequences. Similarly, a map created with hamster single exposure sera positioned BQ.1.1, XBB.1, and BM.1.1.1, a BA.2.75 variant, even further out in antigenic space[24]. Notably, the hamster map did not contain BA.2 sera, which we confirmed to have a big effect on BA.2.75 variant positioning (Supplementary Figs. 12J and 19J). However, the relative position of the overlapping variants correlated in the hamster map with our map. Additionally, a map created from human multi-exposure sera also showed BQ.1.1, XBB, and XBB.1 more distant to BA.2 and BA.4/5[21]. Considering that this study used multi-exposure sera, which are more cross-reactive than single exposure sera and hence generally underestimate antigenic differences and map distances, the position of BQ, XBB, and XBF variants are likely too conservative in our current map. Using single exposure sera to construct an antigenic map has the strong advantage of most accurately representing antigenic relationships among variants because similar titers, and hence proximity in the map, can be attributed to similar neutralization properties.

Our study includes only few single-exposure sera beyond BA.2. Consequently, positioning of some of the new Omicron variants with strong immune escape phenotype is constricted by their low cross-reactivity to the included single-exposure sera, indicated by the high number of titers below detection limit for these variants. Below LOD titers result in antigenic distance estimations that are governed by the fold drop from highest titer in a serum to the LOD (Supplementary Fig. 10C). Although all variants could be positioned using our current approach the map resolution in the area covered by these newer variants is limited. Ideally, a map is constructed using single exposure sera from all variants. This limitation is challenging to overcome using human data, as single exposure sera from BA.2.75, XBB, and BQ.1 variants would be required. However, such sera are not available, as by now nearly everybody has been exposed to SARS-CoV-2 by vaccination, infection, or a combination of both. An alternative could be using sera from very young children after their first infection, but these are difficult to collect and limited in volume. Therefore, aligning antigenic cartography derived from human and animal sera will be important for the future. Considering the substantial escape of BA.2.75, BQ.1, and XBB variants we and others reported in bivalent vaccinated people and our lack of single exposure sera from these variants, their position could be even further away from pre-Omicron and early-Omicron variants than in the here presented map[17-20].

Based on our antigenic map, we constructed antibody landscapes to describe the neutralization profiles of four relevant states of population immunity after immunization with an updated bivalent vaccine: BA.1 biv. or BA.4/5 biv. boosted individuals with or without previous infection. Our findings of substantial immune escape of BA.2.75, BQ.1, and XBB variants after a fourth dose with a bivalent vaccine, despite the discussed limitations of underestimating their antigenic distance to other variants, are in line with previous reports[16,18,19,21]. We found that in hybrid immune individuals with positive N antibody titers a bivalent booster immunization elicited higher overall titers but also enhanced cross-neutralization against these new immune escape variants (Figs. 3 and 4). However, for most individuals in our study the infecting variant is unknown and we therefore do not know if the bivalent booster immunization was the first or second encounter with an Omicron variant spike for these individuals. The increased neutralization magnitude and breadth in hybrid immune people was also observed in the CoVAIL clinical trial analyzing the effects of a fourth dose immunization with different mono- or bivalently updated mRNA vaccines[15]. As the CoVAIL trial recruited individuals in the US in early 2022 when BA.1 was just beginning to circulate the more cross-reactive antibody response compared to uninfected was likely a consequence of infection with a pre-Omicron variant. Considering the clear advantage of hybrid immunity against unexposed variants and independent of the fourth dose vaccine composition, studies delineating differences in immune recall, Ig-antibody composition, and epitope targeting are required to decipher the differences of immune responses to natural SARS-CoV-2 infection and vaccination such that vaccine design can be optimized accordingly.

In a previous study, we found that exposure to three antigenically closely related or to two distinct variants induced broad neutralization across the back-then mapped antigenic space[7]. Our current results and other studies indicate that this does not apply to more recent Omicron lineages. Immune escape variants emerged in areas of antigenic space with low population immunity, and hence extend the antigenic space in that direction. Consequently, additional updates of current COVID-19 vaccines might be needed. For influenza, vaccine composition is annually evaluated and adapted to include variants antigenically representing currently circulating strains. Single exposure antigenic maps and antibody landscapes of more complex exposure histories are combined useful tools for vaccine strain selection. The single exposure antigenic map represents basic antigenic relationships between variants and can highlight antigenic clusters. For Influenza, vaccinating with a strain from one antigenic cluster confers protection against the other cluster strains and historically updates in the vaccine strain composition occurred for variants with 2 grids distance on the antigenic map, i.e., fourfold difference in neutralization[23]. Antibody landscapes, based on a single exposure antigenic map, visualize complex immune profiles after multiple exposures and hence can inform vaccine strain selection by identifying areas of antigenic space with low population immunity and high potential for infection. Further studies on mechanisms of immune escape should be helpful to understand virus evolution and potentially predict strains to be included in future vaccine updates.

An important consideration for updating vaccines is the question whether or not to keep previous, no longer circulating variants in the vaccine composition. For Influenza this is usually not the case, but updated vaccines also back-boosted titers against earlier variants not included in the recent vaccine[22]. The CoVAIL trial found a back-boost of D614G titers after a fourth dose with a variant vaccine that did not contain the ancestral virus strain[25]. Titers against the other vaccine components were lower than against D614G, in line with our results.

Considering the stronger boost of titers against older variants than the more recent vaccine component variant, a vaccination strategy focusing on advanced areas of antigenic space could be a way to build immunity against future SARS-CoV-2 variants with less interference of prior immunity.

## Methods
### Ethics statement
The ethics committee (EC) of the Medical University of Innsbruck has approved the study with EC numbers: 1100/2020, 1111/2020, 1330/2020, 1064/2021, 1093/2021, 1168/2021, 1191/2021, 1197/2021, and 1059/2022. Informed consent has been obtained from study participants.

### Virus culture and sequencing
SARS-CoV-2 isolates (derived from patient material, collected under EC1059/2022) were propagated on Vero cells overexpressing TMPRSS2 and ACE2 receptor. Sequences of isolates were deposited at GISAID (Supplementary Table 2). Vero cells stably overexpressing TMPRSS2 and ACE2 were generated in-house for a previous study[26]. Virus stocks were harvested 48 to 72 h after infection, when a clear cytopathic effect of cells was visible. For sequencing, virus stocks were prepared according

to the Midnight Protocol by ONT (Oxford Nanopore Technologies, Oxford, UK), adapted from Freed et al.[27], and sequenced on an ONT MinION Mk1B sequencer using R9.4.1 flowcells. Sequence analysis was performed using the epi2me-labs/wf-artic[28] nextflow workflow[29], which is based on the ARTIC Network bioinformatics pipeline for SARS-CoV-2[30]. In short, reverse transcription was performed using LunaScript RT SuperMix (NEB, New England Biolabs, Ipswich, MA, USA), followed by PCR to generate tiled amplicons in two primer pools of about 1200 bp length with approximately 20 bp overlap, using Midnight-ONT/V3 primers (ONT) and Q5 HS Master Mix (NEB) (Supplementary Table 4 and[31]). Individual samples were then barcoded utilizing the Rapid Barcoding Kit SQK-RBK110.96 (ONT) and pooled before clean-up with Ampure XP Beads (Beckman Coulter, Brea, CA, USA), addition of sequencing adapter and loading on R9.4.1 flowcells in an Mk1B sequencer (ONT). Following data acquisition on a workstation running Ubuntu 20.04 with MinKNOW (v22.05.5–v22.12.7, ONT), raw read fast5 files were converted to adapter- and barcode-trimmed fastq files, filtered to phred quality score ≥ Q10, using the super high accuracy model of Guppy (ONT, v6.1.5–v6.4.6). The ARTIC Network pipeline for SARS-CoV-2 was pulled from[28] and run using Nextflow (v22.04.4). Here, sequencing reads were filtered to a length between 200 and 1200 bp, aligned to the SARS-CoV-2 reference sequence MN908947.3 using the map-ont preset of minimap2 (v2.18)[32,33], primer sequences soft-trimmed, and resulting bam-files sorted and indexed using samtools (v1.12)[34]. Variant calling used medaka (ONT, v1.5.0) with the r941_min_hac_variant_g507 model. Finally, a consensus sequence was generated using the bcftools consensus module (v1.12)[34] and saved as a FASTA file.

## Plasma samples

Plasma samples were collected from individuals with distinct SARS-CoV-2 exposure histories. We included unvaccinated individuals after infection with ancestral SARS-CoV-2 ($n = 5$), alpha ($n = 10$), beta ($n = 4$), delta ($n = 7$), Omicron BA.1 ($n = 16$), BA.2 ($n = 12$), BA.5 ($n = 1$), or CK.2.1.1 (= BA.5.2.24.2.1.1; $n = 2$) variant, as well as study participants after two ($n = 6$) or three doses ($n = 6$) of BNT162b2 (BNT, Comirnaty, Pfizer/BioNTech) vaccination. Characteristics of those study cohorts have been specified previously[7]. Moreover, plasma samples from individuals after three doses of ancestral SARS-CoV-2 vaccines followed by either ancestral/Omicron BA.1 ($n = 17$) or ancestral/Omicron BA.4/5 ($n = 31$) bivalent boost were analyzed. Patient characteristics, including the number and percentage of female participants (self-reported) for each of the groups analyzed, are listed in Supplementary Table 1.

## ELISA

Bivalent boosted study participants were tested in a SARS-CoV-2 nucleocapsid-specific ELISA (Elecsys®, Anti-SARS-CoV-2, Ref. 09203095, Roche) according to manufacturers' instructions using cobas e411 fully automated analyzer by Roche. Cut-off index (COI) ≥ 1 was considered as positive according to manufactures' instructions.

## Neutralization assay

Neutralization titers of plasma samples were determined against 21 live SARS-CoV-2 isolates (see Supplementary Table 2 and Supplementary Figs. 1–5) performing a focus forming neutralization assay as previously described[7]. Briefly, dilutions (1:16 to 1:16,384) of heat-inactivated plasma samples were incubated with replication competent SARS-CoV-2 for 1 h, before sub-confluent Vero cells overexpressing TMPRSS2 and ACE2 were infected for 2 h. Subsequently, the supernatant was replaced by fresh medium. Cells were fixed further 8 h later using 96% EtOH for 5–10 min. Infected cells were visualized by an immunofluorescence staining (SARS-CoV-2 convalescent serum 1:1000 diluted as primary antibody followed by goat anti-human Alexa Fluor Plus 488-conjugated secondary antibody, 1:1000 diluted; Ref. A48276, Invitrogen, Thermo Fisher Scientific, Vienna, Austria) and counted with an ImmunoSpot S6

Ultra-V reader and CTL analyzer *BioSpot®* 5.0 software (CTL Europe GmbH, Bonn, Germany). Continuous 50% neutralization titers ($IC_{50}$) were calculated by non-linear regression (GraphPad Prism Software 9.0.1, Inc., La Jolla, CA, USA). Titers ≤ 1:1 were set to 1:1 and titers ≥ 1:16,384 were set to 1:16,384. Neutralization titers > 1:16 were considered positive based on previous studies[26].

## Antigenic cartography

Antigenic cartography was first used to analyze seasonal influenza data and has by now been applied to other viruses including SARS-CoV-2[7,21,23,24,35]. Starting from a table of neutralization titers, this titer table is converted into a distance table by calculating the log2 fold change from the maximum titer in a serum to all other titers per serum. Coordinates for each serum and variant pair are then optimized such that their Euclidean distance in the map matches their table distance (Supplementary Fig. 8). A detailed description of the algorithm is given by Smith et al. and the reference page of the Racmacs package[23,36]. Excluding seven sera because of too many titers below the limit of detection, the map was constructed using 98 sera (ancestral virus conv. = 10, alpha/alpha+E484K conv. = 9, beta conv. = 6, delta conv. = 5, BA.1 conv. = 14, BA.2 conv. = 10, BA.5 conv. = 1, CK.2.1.1 conv. = 2, AZ/AZ = 10, AZ/BNT = 10, mRNA1273/mRNA1273 = 10, BNT/BNT = 11, see Supplementary Table 3 for details) in R version 4.2.2[37] and Racmacs version 1.1.35.[36] P.1.1 reactivities were lowered by one two-fold as described previously[7] and 1000 optimization runs were performed with the options "list(ignore_disconnected = TRUE)". Two dimensions were best to represent the data (Supplementary Fig. 7) and map diagnostics showed a good correspondence of map distances and experimental titers (Supplementary Figs. 8). The map was stable to the exclusion of serum groups and number of sera per serum group (Supplementary Figs. 9–20). The ablandscapes package version 1.1.0[38] was used to construct antibody landscape for bivalent BA.1 non-infected (N = 12), infected (N = 5), and bivalent BA.4/5 non-infected (N = 16) and infected (N = 15) with the current map as basis. A single-cone landscape was fit to each individual serum and the GMT landscape calculated from these individual surfaces per serum cohort. Landscapes were fit using the "ablandscape.fit" function with the options "bandwidth = 1, degree = 1, method = " cone," error.sd = 1, control = list(optimize.cone.slope = TRUE)". Geometric mean titers (GMTs) for the multi-exposure groups and fold change calculation were performed using the titertools package version 0.0.0.9001[25], where values below the detection threshold are estimated using a Bayesian approach (described in the Supplement of Wilks et al.[35]).

## Reporting summary

Further information on research design is available in the Nature Portfolio Reporting Summary linked to this article.

# Data availability

Data is available as GitHub repository[39].

# Code availability

The code for the antigenic cartography will be publicly available as GitHub repository[39].

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

## Acknowledgements
We thank Albert Falch, Verena Pittl, Maria Huber, Anna Sauerwein, Anne Heberle, and Toni Rabensteiner for excellent technical and organizational support. We thank Prof. Florian Krammer and Prof. Viviana Simon for sharing their Delta isolate and Prof. Oliver T. Keppler and Dr. Marcel Stern for sharing their Gamma isolate with us. The European Union's Horizon 2020 research and innovation program under grant agreement No. 101016174 and the Austrian Science Fund (FWF) with project number P35159-B supported J.K. The NIH NIAID Centers of Excellence for Influenza Research and Response (CEIRR) contract 75N93021C00014 as part of the SAVE program supported J.K., D.S., and A.N. A.N. was supported by the Gates Cambridge Trust.

## Author contributions
A.R. and A.N. contributed equally to this work. A.R. designed and performed neutralization assays and co-wrote the paper. AN performed all antigenic cartography analyses and wrote the respective sections in the main paper. S.H.W. wrote code. L.K., D.B., and W.B. performed clinical work. Dv.L. co-designed the study. D.J.S. supervised the antigenic cartography and edited the paper. J.K. designed and supervised the study and wrote the paper. All authors edited and reviewed the paper prior to submission.

## Competing interests
D.B. declares to hold stocks of Pfizer and Oxford Nanopore Technologies. The remaining authors declare no competing interests.

 
