## [Peer Review File · Nature Communications]

Characterizing SARS-CoV-2 neutralization profiles after bivalent boosting using antigenic cartographyReviewers' Comments:

Reviewer #1:

Remarks to the Author:

This manuscript describes the live virus neutralization sensitivity of more recently circulating Omicron variants against single variant infected, double or triple vaccine boosted or hybrid immune individuals and uses this data to develop antigenic cartography maps and/or antibody landscapes. The antigenic map clearly highlights the segregation between pre-Omicron and Omicron lineages as previous studies have shown, and further defines the antigenic differences between more recently circulating BA.2, BA.5 and XBB recombinant variants. In addition, the BA.5 variants cluster somewhat with the initial BA.5 and the XBF.3 variant is the most antigenically distinct of the Omicron variants. However, the BA.2 variants, clustered more closely with the BA.5 variants rather than with the initial BA.2, likely due to low levels of cross-neutralization of these BA.2 variants by all sera used in the analysis. Lastly, the antibody landscapes created using sera from individuals who had received either a BA.1 or BA.4/5 bivalent booster and then had either experienced or not experienced a breakthrough infection showed that individuals with hybrid immunity developed higher, more cross-reactive titres, consistent with previous studies looking at hybrid immunity following either primary series or bivalent booster vaccinations.

Overall the study is clearly written and will be of general interest. However the authors should comment on the validity of using single variant infected sera (from either humans or animals) to inform vaccine composition for use in adults, given that that this immune scenario is now extremely rare. I have noted several places where this is relevant below

I would recommend that the manuscript is accepted, subject to minor revisions below:

Abstract

Line 18 and throughout the manuscript - "Omicron" should be capitalised.

Line 26 - remove the word "omicron"

"...ancestral virus and either BA.1 or BA.4/5 omicron"

Introduction

Line 36 - remove the word "type"

"...coronavirus type 2 (SARS-CoV-2)..."

Line 36 - replace "evolved" with "emerged"

"...coronavirus type 2 (SARS-CoV-2 have evolved..."

Line 36 - replace summer a time frame eg "beginning of 2022", as season is dependent on the hemisphere.

"...end of 2021 and summer 2022..."

Line 65 - replace "being dominant" with "dominating"

"...different lineages being dominated..."

Line 65 - replace "the initial" with "parental" AND remove the work "variant"

"...However, BA.1 and the initial BA.5 variant..."

Line 71 - Based on my interpretation of this and all influenza antigenic cartography, it seems that while the method of using mostly single variant/strain exposed sera may be less complex to understand, it may be a contributor to vaccine strain mismatching. Given that the majority of the global population has experienced multiple prior infections, I would argue that this kind of analysis is

no longer essential or even biologically relevant for adult populations, though it may inform vaccine choices for immunologically naïve individuals like children.

Therefore, I would recommend tempering this statement by using the word "informative" rather than "essential." This is a recurring topic, as described above.

Line 75 - include an "a" before the word "basis"
"....antigenic map as basis..."

Methods

Line 113 – how was the positivity cut-off of greater than or equal to 1:16 determined?

A paragraph should be included describing both the virus culture and third-generation sequencing methods used in this paper.

Results

Line 155 to 157, sentence beginning "For generation of reliable antigenic maps..."

Here again, I would challenge the authors to comment on the impacts of using single variant/strain exposed sera in both SARS-CoV-2 and influenza antigenic cartography analyses. Do these "reliable" maps have relevance and therefore contribute to accurate vaccine strain/variant selection in the context of the complex prior infection histories?

Line 163 – replace "newly" with "recently"
"...against our panel of newly isolated omicron..."

Line 165 - the three times BNT vaccinated individuals were not included in the antigenic map creation, but were used as a reference, however is it possible to include a map where these were used in the supplementary files. It would be interesting to see how compressed the distances between Omicron variants become, if at all. If this is not possible, could the authors provide an explanation in the text, either here or in line 185 of the results section?

Line 185 and Figure 2

Is it possible to show a zoomed in view (as an inset for example) of Omicron variants alone, ie excluding the pre-Omicron variants, to get a more high resolution visualisation of the antigenic differences within the pre-Omicron group?

Line 190 – replace the word "phenotypic" with "phenotypically"
"...hence map distance reflects phenotypic distinct antigenic relationships."

Line 236

The authors should mention that timing between breakthrough infection and sampling could also contribute to the distance between hybrid immune and vaccine only individuals in the BA.1 versus BA.4/5 groups.

Discussion

Line 259 – Supplementary Figure X is referenced here, please correct or remove

Line 260 – The hamster antigenic map from a previous study is referenced here. For ease of reference, it might be good to have this map in the supplementary figures.

Line 264 – Again here the authors comment on how the antigenic differences between BQ,XBB and XBF variants are too conservative given the use of multi-exposure sera, and again I would say that these "conservative" estimates are likely what is more relevant globally. I would urge the authors to address this here again in the discussion.

Reviewer #2:

Remarks to the Author:

In this study, Rössler et al. conducted antigenic analyses on emerging variants BA.2.75, BF.7, BQ, XBB, and XBF using human plasma collected from a cohort of individuals infected or vaccinated with COVID vaccines. Additionally, a neutralization study was performed on a cohort of individuals who received three doses of the ancestral variant vaccine, followed by a fourth dose of either a bivalent ancestral+BA.1 or ancestral+BA.4/5 vaccine. The researchers observed variations in neutralizing titers against all variants among individuals without N antibodies. The authors claimed evidence of immune escape in bivalently boosted individuals for the BA.2.75, BF.7, BQ, XBB, and XBF variants.

Major comments:

1) The antigenic cartography does not seem to support significant antigenic changes for BA.2.75, BF.7, BQ, XBB and XBF variants. Antigenic distances were too small.

2) The data from this study does not strongly support the claim of immune escape in BA.2.75, BF.7, BQ, XBB, and XBF variants. As shown in Figure 4, there were notable variations in neutralization titers among these individuals without N antibodies. However, these variations could be attributed to the diversity of immune responses among individuals, especially considering the influence of pre-existing immunity resulting from prior vaccination.

3) This study primarily consists of observational analysis and does not offer a clear explanation regarding the antigenic changes observed in BA.2.75, BF.7, BQ, XBB, and XBF variants, as well as their association with immune escape, which this study claimed.

4) It would be valuable to provide the infection history of individuals who tested positive for N antibodies, indicating the variant(s) associated with previous infection(s). This information would help determine whether these individuals were infected with pre-Omicron variants or Omicron variants. Though, it is worth noting that hybrid immunity, which combines natural infection and vaccination, has been well-documented in increasing neutralizing titers.

Reviewer #3:

None

Reviewer #1 (Remarks to the Author):

This manuscript describes the live virus neutralization sensitivity of more recently circulating Omicron variants against single variant infected, double or triple vaccine boosted or hybrid immune individuals and uses this data to develop antigenic cartography maps and/or antibody landscapes. The antigenic map clearly highlights the segregation between pre-Omicron and Omicron lineages as previous studies have shown, and further defines the antigenic differences between more recently circulating BA.2, BA.5 and XBB recombinant variants. In addition, the BA.5 variants cluster somewhat with the initial BA.5 and the XBF.3 variant is the most antigenically distinct of the Omicron variants. However, the BA.2 variants, clustered more closely with the BA.5 variants rather than with the initial BA.2, likely due to low levels of cross-neutralization of these BA.2 variants by all sera used in the analysis. Lastly, the antibody landscapes created using sera from individuals who had received either a BA.1 or BA.4/5 bivalent booster and then had either experienced or not experienced a breakthrough infection showed that individuals with hybrid immunity developed higher, more cross-reactive titres, consistent with previous studies looking at hybrid immunity following either primary series or bivalent booster vaccinations.

Overall the study is clearly written and will be of general interest. However the authors should comment on the validity of using single variant infected sera (from either humans or animals) to inform vaccine composition for use in adults, given that that this immune scenario is now extremely rare. I have noted several places where this is relevant below

We have addressed this point as response to the reviewer's comments and in the revised version of the manuscript in lines 72-81, 178-181, 189-195, 311--314, and 356-366. Please see responses below for more detailed comments. Briefly, we consider that single exposure sera reflect the basic antigenic differences among variants best as map distances are based on fold drops in neutralizing antibody titers between variants and multiple exposures increase cross-neutralization also to non-exposed variants and will consequently reduce these fold drops and map distances. We agree with the reviewer that single exposed individuals are very rare by now but that most individuals have more complex exposure histories. To visualize these we constructed antibody landscapes based on the single exposure map. We consider that the combination of a single exposure map to identify antigenic clusters and antibody landscapes of more complex and population relevant immune scenarios with this map as a basis to identify areas with low population immunity should be most helpful for vaccine strain selection.

I would recommend that the manuscript is accepted, subject to minor revisions below:

Abstract

Line 18 and throughout the manuscript - "Omicron" should be capitalised.

changed

Line 26 - remove the word "omicron"

"...ancestral virus and either BA.1 or BA.4/5 omicron"

changed

Introduction

Line 36 - remove the word "type"

“...coronavirus type 2 (SARS-CoV-2)...”

changed

Line 36 - replace “evolved” with “emerged”

“...coronavirus type 2 (SARS-CoV-2 have evolved...”

changed

Line 36 – replace summer a time frame eg “beginning of 2022”, as season is dependent on the hemisphere.

“...end of 2021 and summer 2022..”

We exchanged “summer” by “middle of”

Line 65 – replace “being dominant” with “dominating”

“...different lineages being dominated..”

changed

Line 65 – replace “the initial” with “parental” AND remove the work “variant”

“...However, BA.1 and the initial BA.5 variant...”

changed

Line 71 – Based on my interpretation of this and all influenza antigenic cartography, it seems that while the method of using mostly single variant/strain exposed sera may be less complex to understand, it may be a contributor to vaccine strain mismatching. Given that the majority of the global population has experienced multiple prior infections, I would argue that this kind of analysis is no longer essential or even biologically relevant for adult populations, though it may inform vaccine choices for immunologically naïve individuals like children.

Therefore, I would recommend tempering this statement by using the word “informative” rather than “essential.” This is a recurring topic, as described above.

We agree with the reviewer that considering current population immunity single exposure sera are not that relevant for vaccine strain selection. We thank the reviewer for pointing out that this was not stated clearly enough in the manuscript. However, for accurately representing antigenic relationships of virus variants in an antigenic map, single exposure sera are essential. The reason for this is that a map is constructed based on fold drops from the maximum titer antigen in a serum (usually the infecting/vaccinating antigen) to other variant titers. Here, single exposure is essential because multiple exposures could lead to deflated fold drops from one variant to the other because the individual has been exposed to both variants, rather than because exposure to one variant elicits high titers against the other variant and hence indicates that these variants are antigenically similar. Using single exposure sera to construct an antigenic map allows us to control for this. We thus argue that, to determine the most fundamental antigenic differences among variants, and to represent these in an antigenic map, single exposure sera are essential.

We fully agree with the reviewer, that using such a map alone for vaccine strain selection is only part of the process and that additionally considering multi exposure sera is essential. For this reason, antibody landscapes were developed which allow to combine the antigenic relationships inferred from single exposure sera with immunity after multiple exposures. Here, the slope and height of the landscape visualize the neutralization profile in antigenic space and the landscape shows which variants escape multi exposure immunity. Still, to construct an antibody landscape an antigenic map as basis is required.

We have changed the statement on line 72 to make the above described clearer:

“However, to clearly disentangle antigenic relations between virus variants, single exposure sera are most informative as antigenic maps represent titer differences among variants and are constructed based on fold drops from the maximum titer antigen in a serum (usually the infecting/vaccinating antigen) to other variant titers. We have previously shown that multiple exposures decrease these fold drops in titer between exposed and non-exposed variants. Using multi exposure sera to construct a map therefore may underestimate distances between variants due to higher cross-neutralization after multiple exposure rather than similar titers due to similar neutralization properties. Antibody landscapes, which are constructed to visualize neutralization profiles in a third dimension on top of a single exposure antigenic map, are a more appropriate tool to depict immunity after more complex exposure histories.”

Line 75 - include an “a” before the word “basis”

“.....antigenic map as basis...”

changed

Methods

Line 113 – how was the positivity cut-off of greater than or equal to 1:16 determined?

We used the cut-off of greater than 1:16 based on previous studies and now added a reference to this in the Methods section, see lines 134-135.

A paragraph should be included describing both the virus culture and third-generation sequencing methods used in this paper.

We have added paragraphs describing virus culture and sequencing to the Methods section, see lines 97-103.

Results

Line 155 to 157, sentence beginning “For generation of reliable antigenic maps...”

Here again, I would challenge the authors to comment on the impacts of using single variant/strain exposed sera in both SARS-CoV-2 and influenza antigenic cartography analyses. Do these “reliable” maps have relevance and therefore contribute to accurate vaccine strain/variant selection in the context of the complex prior infection histories?

As this relates to the reviewer’s comment in the introduction Line 71, we would like to point towards our response to the previous comment for a more detailed answer. Briefly, single exposure maps most

accurately represent antigenic relationships (neutralization properties) across variants because these controlled exposure scenarios allow us to infer that similar titers against two variants in the same sera are due to similar antigenic properties rather than exposure to the two variants, or exposure to two variants that are antigenically related to the titrated variants.

To make this point clearer we now changed “reliable antigenic maps” into “antigenic maps that reflect the basic antigenic relationships among variant”, see lines 178-179.

We agree that multiple prior infections can modify the immune profile in complex ways, which is why antibody landscapes are an important tool to investigate this and identify areas (variants) where population immunity is low. As single exposure maps are the basis for these landscapes, they are highly relevant for vaccine strain selection. Additionally, single exposure maps allow one to identify clusters of antigenically similar strains, and vaccination with one of these cluster strains will likely induce protection against the other strains. For vaccine strain selection a combination of a single exposure antigenic map that is used to construct antibody landscapes of sera representing more complex forms of population immunity should be most useful.

In response to this comment, we have extended the benefit of single exposure sera maps in the discussion, lines 311-314 and 356-366.

Line 163 – replace “newly” with “recently”

“...against our panel of newly isolated omicron...”

changed

Line 165 - the three times BNT vaccinated individuals were not included in the antigenic map creation, but were used as a reference, however is it possible to include a map where these were used in the supplementary files. It would be interesting to see how compressed the distances between Omicron variants become, if at all. If this is not possible, could the authors provide an explanation in the text, either here or in line 185 of the results section?

We have not included the three times BNT vaccinated individuals in the construction of the antigenic map as we saw in a previous study that three exposures to antigenically close variants, i.e. three doses of BNT vaccination, induced good cross-neutralization to non-exposed variants and consequently a lower fold drop in neutralizing titers between exposed and non-exposed variants (Rössler et al. 2022, Nature Communications). Neutralization profiles of the three dose vaccinated individuals were more similar to individuals after multiple infections than to ancestral variant single exposure sera. As outlined above, we do not recommend the use of multi exposure sera in antigenic map construction. But we do agree with reviewer that visualizing the neutralization profiles of the 3xBNT cohort is an interesting point. As we believe that antibody landscapes are the more appropriate way to illustrate neutralization profiles after complex exposure histories, as we now also outlined in the manuscript, see lines 72-81. We consequently added antibody landscapes for this cohort as new Supplementary Figure 23. See also lines 189-195 in the updated results section.

“Three times BNT vaccinated individuals (BNT/BNT/BNT) were included as reference but the data from this cohort has not been used for calculation of the antigenic map as multi exposure sera likely underestimate antigenic relationships due to increased cross-reactivity and we previously showed that neutralization profiles of three dose vaccinated individuals were more similar to those of individuals after re-infection with an antigenically distinct variant than after single infection. However, we

constructed antibody landscapes for this group to visualize neutralization profiles (Supplementary Figure 23).”

Line 185 and Figure 2

Is it possible to show a zoomed in view (as an inset for example) of Omicron variants alone, ie excluding the pre-Omicron variants, to get a more high resolution visualisation of the antigenic differences within the pre-Omicron group?

We agree with the reviewer that in theory it is possible and can be advantageous to zoom in on regions of an antigenic map, as we did in Figure 2 compared to the non-zoomed in map version in Supplementary Figure 6. However, as relationships and distances in an antigenic map are relative to the variants and sera in it, and the resolution of the Omicron area of antigenic space is low due to few single exposure sera in that area of antigenic space, we prefer to not show this region without the other variants. It might give the impression of a lot of certainty for the positions of the Omicron variants, whereas we have shown in our analysis in Supplementary Figure 16 that the resolution for their position is limited.

Line 190 – replace the word “phenotypic” with “phenotypically”

“...hence map distance reflects phenotypic distinct antigenic relationships.”

changed

Line 236

The authors should mention that timing between breakthrough infection and sampling could also contribute to the distance between hybrid immune and vaccine only individuals in the BA.1 versus BA.4/5 groups.

We added a comment addressing this, see lines 274-276.

Discussion

Line 259 – Supplementary Figure X is referenced here, please correct or remove

We thank the reviewer for their careful reading of the manuscript and pointing our mistake out. We corrected it to Supplementary Figures 12 and 19.

Line 260 – The hamster antigenic map from a previous study is referenced here. For ease of reference, it might be good to have this map in the supplementary figures.

As the map mentioned here has been published by another group in a different journal (Lancet Microbe) we did not include the Figure here but only the reference.

Line 264 – Again here the authors comment on how the antigenic differences between BQ,XBB and XBF variants are too conservative given the use of multi-exposure sera, and again I would say that these “conservative” estimates are likely what is more relevant globally. I would urge the authors to address this here again in the discussion.

As outlined in response to the comments to lines 71 and 155-157 of the originally submitted manuscript version (see above), we believe that a combination of a single exposure antigenic map to disentangle basic antigenic relations between variants and antibody landscapes calculated on top of this map to represent neutralization profiles after more complex exposure histories is beneficial. We addressed this point in the revised version of the manuscript in the Introduction, Results and Discussion, see lines 72-81, 178-181, 189-195, 311--314, and 356-366.

Reviewer #2 (Remarks to the Author):

In this study, Rössler et al. conducted antigenic analyses on emerging variants BA.2.75, BF.7, BQ, XBB, and XBF using human plasma collected from a cohort of individuals infected or vaccinated with COVID vaccines. Additionally, a neutralization study was performed on a cohort of individuals who received three doses of the ancestral variant vaccine, followed by a fourth dose of either a bivalent ancestral+BA.1 or ancestral+BA.4/5 vaccine. The researchers observed variations in neutralizing titers against all variants among individuals without N antibodies. The authors claimed evidence of immune escape in bivalently boosted individuals for the BA.2.75, BF.7, BQ, XBB, and XBF variants.

Major comments:

1) The antigenic cartography does not seem to support significant antigenic changes for BA.2.75, BF.7, BQ, XBB and XBF variants. Antigenic distances were too small.

We find that most omicron variants, both parental as well as newly emerged omicron variants, form a new cluster that is distinct from the pre-omicron cluster. Some BA.5 variants, i.e. BA.5.3.2, BA.5.2.1, BE.1.1 and BF.7, do not differ substantially in spike sequence and according to our titrations in neutralization profiles and are therefore located within one antigenic unit of each other on the map as expected. BQ.1.1 and all BA.2 based newly emerged variants (XBB.1, XBB.1.5.1, XBF.3, CB.1, CH.1.1, BR.3) are located further away from these variants, many with more than one antigenic unit difference. It is especially interesting that all these BA.2-derived variants are located far away from BA.2, e.g. the distance between BA.2 and delta is smaller than the distance between BA.2 and BA.2-derived variants. We added some additional comments on this in the results section, see lines 221-230.

We agree with the reviewer that we might underestimate antigenic distances for some of the newly emerged variants. One major limitation in antigenic cartography based on human sera can be attributed to the lack of human first exposure sera against BA.2.75, BQ, XBB and XBF variants. These variants strongly escape neutralization by the human first exposure sera we could include in our map and consequently many neutralizing antibody titers were below the limit of detection (LOD). As map distances correspond to titer fold changes, the antigenic differences for a variant with detectable titers and a variant with titers below LOD present a degree of uncertainty, as the assay does not determine how much below the LOD the titer against the second variant is. Although antigenic cartography deals with this uncertainty by allowing distances with one titer <LOD to be larger or equal to the fold drop from the detectable titer to the LOD, it can result in an underestimation of antigenic distances. Indeed, our analyses on the effect of measurement uncertainty, and more precisely titer magnitude, in Supplementary Figure 10C show the effect of varying detectable titers on the position of the recent variants. Hence, the small distances in this antigenic map can be attributed to low detectable titers in the human sera and <LOD titers against the recent Omicron variants.

Note, hamster sera antigenic maps, that have sera against some of the more recent variants do place the more recent variants more distant from the early omicrons. One does need to be careful however to fully trust hamster sera as testing how well hamster sera data represented human sera is still a work-in-progress.

We hope to have addressed this limitation more extensively in the revised manuscript by changing the following in the results and discussion, see lines 277-288 and 315-321.

“Considering the almost complete escape from single exposure sera of the BQ, XBF and BA.2.75 variants, their position in antigenic space could be even further away from pre-Omicron and early Omicron variants than in the current map. As many titers against these variants were below the LOD (limit of detection), the actual titer difference is censored by the LOD. Consequently, a variant may be positioned at a distance corresponding to the fold drop from maximum titer to the LOD, however, it cannot accurately be estimated how much further away the variant actually is (Supplementary Figure 10C, 16). This could only be resolved by increasing the resolution in this area of antigenic space through addition of first exposure sera from these variants, which are challenging to obtain given the current situation of population immunity. To test the impact of the underestimation of antigenic escape of the recent Omicron variants on the antibody landscapes, we constructed landscapes where the new variants were not included in the fitting procedure (Supplementary Figure 28).”

“Our study includes only few single-exposure sera beyond BA.2. Consequently, positioning of some of the new Omicron variants with strong immune escape phenotype is constricted by their low cross-reactivity to the included single-exposure sera, indicated by the high number of titers below detection limit for these variants. Below LOD titers result in antigenic distance estimations that are governed by the fold drop from highest titer in a serum to the LOD (Supplementary Figure 10C). Although all variants could be positioned using our current approach the map resolution in the area covered by these newer variants is limited.”

2) The data from this study does not strongly support the claim of immune escape in BA.2.75, BF.7, BQ, XBB, and XBF variants. As shown in Figure 4, there were notable variations in neutralization titers among these individuals without N antibodies. However, these variations could be attributed to the diversity of immune responses among individuals, especially considering the influence of pre-existing immunity resulting from prior vaccination.

The reviewer is correct that in Figure 4 there is some variation in titers among the individuals. However, when comparing titer changes between variants for individual patients we see similar patterns, i.e. drop of neutralizing titers for BA.2.75, BQ, XBB and XBF variants relative to pre-omicron or early omicron variants. We generally do not see considerable drop in titers against BF.7 compared to parental BA.5, as also expected given the high similarity in spike for the two viruses. We reference to this in the results section, see lines 224-226. In Figure 4 titers from each participant against different variants are connected by lines. As the figure might be busy due to different titers against the reference variants among individuals, we additionally calculated fold changes in titers for the variants included in the vaccines, i.e. ancestral, BA.1 and BA.5 against each virus variant. Titers against BA.2.75, BQ, XBB and XBF variants dropped depending on the reference variant ~4-fold or more. Please see Supplementary Figure 26 and graph below, respectively. We added now a clearer reference to this Supplementary Figure in the results section, see lines 263-266.

3) This study primarily consists of observational analysis and does not offer a clear explanation regarding the antigenic changes observed in BA.2.75, BF.7, BQ, XBB, and XBF variants, as well as their association with immune escape, which this study claimed.

We agree with the reviewer that our study is mainly observational as both antigenic maps and antibody landscapes are methods to visualize data. The aim of our study was to characterize immune scenarios which are relevant from a public health perspective against recent variants, not offer mechanistic explanations, which we hoped to make clear in the title of our manuscript “Characterizing SARS-CoV-2 neutralization profiles after bivalent boosting using antigenic cartography” and elaborated in the abstract. To disentangle mechanisms of immune escape a different experimental setup would be required. However, we also agree with the reviewer that further studies on mechanisms of immune escape will be most helpful to understand virus evolution and potentially predict strains to be included in future vaccine updates and added a comment on this in the discussion, see lines 364-366.

We have checked the manuscript to see how much we are claiming beyond observation, and do not think that we have claimed beyond observational analyses. We do make claims that a benefit of our study is to help with vaccine strain selection for an updated COVID-19 vaccine. Here, the combination of describing basic antigenic relations of new variants via single exposure antigenic maps and antibody landscapes to identify holes in population immunity should be most helpful. Please note our data, maps, and landscapes have been sought out for presentation at many international fora related to SARS-CoV-2 virus evolution and vaccine strain selection, which we write here in support of our claim of (at least) interest for these questions. We added a paragraph addressing this also in regard with updating Influenza vaccines to the discussion, see lines 356-366.

4) It would be valuable to provide the infection history of individuals who tested positive for N antibodies, indicating the variant(s) associated with previous infection(s). This information would help determine whether these individuals were infected with pre-Omicron variants or Omicron variants. Though, it is worth noting that hybrid immunity, which combines natural infection and vaccination, has been well-documented in increasing neutralizing titers.

We agree with the reviewer that it would be very informative to know the infecting variant for the N positive individuals and especially if this has been a pre-omicron or an omicron variant. Unfortunately most of the N positive individuals had no previous positive PCR or antigen test. Only 4 of the total of 20 N positive (5 BA.1 biv. and 15 BA.4/5 biv.) had a previous positive PCR, 1 likely infected with the alpha variant and 3 likely with BA.2. We have depicted this information in Supplementary Table 1. We now also added the information about the infecting variant for the 4 individuals with known infection history to the text and the information that this information is unfortunately unknown for the rest, see lines 245-246 and 249-250.

Reviewers' Comments:

Reviewer #2:

Remarks to the Author:

The revised manuscript has indeed experienced notable improvements when compared to the previous version. A remaining limitation is the ongoing lack of clarity regarding the implications of these minor antigenic changes observed among the Omicron variants.

Reviewer #3:

Remarks to the Author:

I thank the authors for their thoughtful and detailed responses to our critiques. The authors have sufficiently addressed both the minor and major comments that I raised. I support the publication of this manuscript.